# Long-Term Outcomes after Surgery for Pheochromocytoma and Sympathetic Paraganglioma

**DOI:** 10.3390/cancers15112890

**Published:** 2023-05-24

**Authors:** Francesca Torresan, Arianna Beber, Donatella Schiavone, Stefania Zovato, Francesca Galuppini, Filippo Crimì, Filippo Ceccato, Maurizio Iacobone

**Affiliations:** 1Endocrine Surgery Unit, Department of Surgery, Oncology and Gastroenterology, University of Padova, 35128 Padova, Italy; francesca.torresan@unipd.it (F.T.); arianna.beber@gmail.com (A.B.); schiavone.donatella@gmail.com (D.S.); 2Familial Cancer Clinic, Veneto Institute of Cancer, Istituto Oncologico Veneto IOV IRCCS, 35128 Padova, Italy; stefania.zovato@iov.veneto.it; 3Pathology Unit, Department of Medicine, University of Padova, 35128 Padova, Italy; francesca.galuppini@unipd.it; 4Radiology Unit, Department of Medicine, University of Padova, 35128 Padova, Italy; filippo.crimi@unipd.it; 5Endocrinology Unit, Department of Medicine, University of Padova, 35128 Padova, Italy; filippo.ceccato@unipd.it

**Keywords:** pheochromocytoma/paraganglioma prognosis, survival, adrenalectomy

## Abstract

**Simple Summary:**

Pheochromocytoma and sympathetic paraganglioma (PHEO/sPGL) are rare and mainly sporadic tumors arising from chromaffin cells, for which hereditary variants occur in about one-third of cases. The discrimination between benign and malignant lesions is extremely challenging in PHEO/sPGL and the prognosis is indeterminate. The study aimed to analyze the long-term outcomes of PHEO/sPGL in a cohort of 170 patients. The results demonstrated that new tumor recurrence occurred more frequently with hereditary variants, even though new tumors also reoccurred with sporadic variants. Moreover, even if the risk of metastatic recurrence was higher with malignant variants at diagnosis, the risk also occurred with apparently benign cases. This suggests that lifelong follow-up is required for PHEO/sPGL, even in sporadic and apparently benign cases.

**Abstract:**

Background: The prognosis of pheochromocytoma and sympathetic paraganglioma (PHEO/sPGL) is difficult to predict at the time of diagnosis and long-term follow-up data are scarce, especially for apparently benign and sporadic variants. The aim of the study was to analyze the long-term outcomes in PHEO/sPGL patients. Methods: A monocentric series of 170 patients who underwent surgery for PHEO/sPGL was analyzed. Results: The study cohort included 91 female and 79 males with a median age of 48 years (range 6–83). The majority of PHEO/sPGL cases were considered apparently benign at the time of diagnosis; evident malignant behavior was found in 5% of cases. The overall 10-year risk of recurrence was 13%, but it rose up to 33% at 30 years. The risk of new tumor recurrence was higher in patients with hereditary tumors, but the risk was still significant in patients with apparently sporadic variants (20-year risk: 38% vs. 6.5%, respectively; *p* < 0.0001). The risk of metastatic recurrence was higher in patients with locally aggressive tumors at diagnosis, but the risk was present also in apparently benign variants (5-year risk: 100% vs. 1%, respectively; *p* < 0.0001). Conclusions: Lifelong follow-up is required not only for hereditary PHEO/sPGL but also for apparently benign and sporadic tumors at diagnosis because of the risk of long-term recurrent disease.

## 1. Introduction

Pheochromocytoma (PHEO) and sympathetic extra-adrenal abdominal paraganglioma (sPGL) are rare neuroendocrine tumors arising from chromaffin cells of the adrenal medulla or extra-adrenal sympathetic paraganglia, respectively.

They are usually catecholamine-secreting tumors, occurring with signs and symptoms of hemodynamic and metabolic instability, such as sustained or paroxysmal hypertension [1]. Even if most PHEO/sPGL are sporadic, approximately one-third of cases are caused by specific germline pathogenic variants with variable penetrance and clinical expressivity. Several genes have been discovered in the pathogenesis of these tumors, and several syndromes have been described [2,3,4].

The histopathological discrimination between benign and malignant lesions is extremely challenging. The presence of metastases in organs that do not naturally contain chromaffin tissue is the only certain way to diagnose malignancy; therefore, in 2017, the World Health Organization (WHO) renamed the “malignant” nomenclature as “metastatic” [5], and the terms “benign” and “malignant” were abandoned. The most recent 2022 WHO classification suggests considering PHEO as intra-adrenal sPGL and treating all tumors as malignant lesions with variable metastatic potential [6,7]. Moreover, a novel staging system for PHEO/sPGL has been introduced by the American American Joint Committee on Cancer (AJCC) [8].

After surgery, most patients with PHEO/sPGL are usually considered definitively cured since they remain disease-free for a long time; thus, in the absence of evident pathological features of malignancy at the time of diagnosis, the follow-up is usually discontinued. However, recurrences may manifest as new tumor development in chromaffin tissues (usually the contralateral adrenal gland or other sympathetic paraganglia, as commonly occurs with hereditary variants) or as distant metastases [9]. Recent studies estimated a 5-year probability of recurring disease at 6.5% after focused clinical, laboratory, and radiological follow-up and disease relapse in one-third of patients more than 10 years after initial surgery [10,11]. Therefore, lifelong follow-up should be performed.

However, due to the low prevalence of disease and, consequently, the scarcity and even lack of long-term follow-up data, the prognosis of PHEO/sPGL is difficult to predict at the time of diagnosis, especially for apparently benign and sporadic PHEO/sPGL. Furthermore, the complexity of the disease and the continuous discovery of new susceptibility genes require more studies.

The aim of the study was to analyze the main features and long-term outcomes in terms of survival and recurrence (metastatic recurrence and/or new tumors) in a large monocentric series of PHEO/sPGL patients.

## 2. Materials and Methods

The present retrospective study included a series of 170 patients who underwent surgery for PHEO/sPGL at the Endocrine Surgery Unit of Padova University Hospital, Italy, between 1979 and 2019, and the study was approved by the institutional review board. In- and outpatient medical records were reviewed to gather relevant demographics (gender, age at surgery), clinical, and biochemical features, genetic results, surgical treatment, and pathology findings. The follow-up data included clinical, biochemical, and radiological evaluations. The preoperative and postoperative workup included 24 h urinary catecholamine, metanephrine, and dopamine levels, abdominal computed tomography (CT) scan or magnetic resonance imaging (MRI), and in some cases, 123I-metaiodobenzylguanidine (MIBG) scintigraphy or 18F-dihydroxyphenylalanine (DOPA)-Positron Emission Tomography (PET) and/or 18F-fluorodeoxyglucose (FDG)-PET scan.

Genetic testing for germline pathogenic variants predisposing to hereditary PHEO/sPGL was performed after genetic counselling. Written informed consent was obtained and DNA was extracted from peripheral blood samples. Genetic analysis was performed using next-generation sequencing (NGS) technology; the NGS sequencing analysis included *FH*, *MAX*, *NF1*, *RET*, *SDHA*, *SDHAF2*, *SDHB*, *SDHC*, *SDHD*, *TMEM127*, and *VHL*.

Surgery included unilateral or bilateral laparoscopic or laparotomic/lombotomic approaches. In the pathology reports, the Pheochromocytoma of the Adrenal gland Scaled Score (PASS) was available in some cases. As previously described, a PASS score > 4 was considered with a potential for biologically aggressive behavior [12]. Postoperative outcomes and follow-up data were assessed according to the last available medical records, including clinical, laboratory, and imaging reassessment. Only patients with complete outcome follow-up data (at least 24 months) were included for survival and recurrence rate analysis. The final end points of the study were the long-term outcomes in terms of survival and recurrence. At diagnosis, PHEO/sPGL were classified according to the 2017 AJCC staging system [8]. Tumors at stage I (including PHEO < 5 cm without extra-adrenal invasion, nodal or distant metastases) and stage II (including PHEO > 5 cm and sPGL of any size without extra-adrenal/tumoral invasion, nodal or distant metastases) were considered as apparently benign lesions with indeterminate behavior. Tumors at stage III (i.e.,: any size with invasion into surrounding tissues and/or regional lymph node metastases) and stage IV (i.e., with distant metastases) were considered as malignant. Remission was defined by the normalization of laboratory data in the absence of signs of disease relapse during imaging assessment for a period of at least 6 months after surgery. Recurrence was defined as disease relapse after postoperative remission and further classified as “new tumor recurrence” in case of relapse in organs containing chromaffin tissue or “metastatic recurrence” in case of relapse in organs where chromaffin tissue is normally absent. Survival was calculated as the time from surgery until death or the last available follow-up. Only disease-specific causes of death were considered for mortality rate analysis.

Data were expressed as absolute numbers, percentages, mean, median, range, standard deviation (SD), or 95% confidence interval (CI). The Student’s *t*-test, Mann–Whitney, Chi-square, and Fisher’s exact tests were appropriately used. Positive predictive value (PPV) and negative predictive value (NPV) for PASS score were defined by the ratio of true positive/(true positive + false positive) and true negative/(true negative + false negative), respectively. Survival data were analyzed by the Kaplan–Meier method and the log-rank test. Recurrence was calculated as the time from surgery until the time of disease relapse and analyzed by the Kaplan–Meier method and the log-rank test. Univariate and multivariate analyses of factors affecting recurrence were conducted by logistic regression and used the Cox proportional hazards regression model, respectively.

A *p*-value < 0.05 was considered statistically significant.

## 3. Results

The clinical characteristics of the 170 patients are summarized in Table 1.

The study included 91 female and 79 males; the median age at surgery was 48 years (range 6–83). A PHEO was found in 155 patients (unilateral in 90%, bilateral in 10%), a single sPGL was found in 10 patients, and a combined PHEO/sPGL was found in 5 patients (1 patient with a bilateral PHEO and a synchronous abdominal sPGL, 2 patients with a bilateral PHEO and a metachronous abdominal sPGL, 1 patient with a neck PGL and a metachronous PHEO, and 1 patient with a unilateral PHEO and a metachronous sPGL).

The majority of patients (59% of cases) were referred to the outpatient clinic for symptoms related to arterial hypertensive spikes, while 10% of cases were referred for lumbar or abdominal pain, and a less frequent 4% of cases were referred for a cardio or cerebrovascular emergency (Tako-Tsubo syndrome, stroke). The mass was incidentally discovered in 27% of cases. Catecholamine hypersecretion was found in 162 (95%) cases, while the remaining 8 tumors (5%) were presumably non-secreting.

The majority of patients (95%) had radiological, intraoperative, and pathological characteristics indicating indeterminate and apparently benign PHEO/sPGL (AJCC stage I *n* = 100, stage II *n* = 61) at the time of diagnosis, while malignant behavior (main vessel infiltration, gross adjacent tissue invasion or nodal metastases; AJCC stage III) was found in 5% of cases. None of the patients had distant metastases at diagnosis (AJCC stage IV).

In the pathology data, the PASS score was available for 46 PHEO. In 9 cases (20%) the PASS score was ≥4; in 3 of these patients, a malignant AJCC stage III PHEO was found at diagnosis. In 37 (80%) cases, the PASS score was <4 and none of these patients developed metastatic recurrence during follow-up (PPV 33%; NPV 100%).

### 3.1. Genetic Analysis

The analysis of germline pathogenic variants was available for 107 patients: pathogenic variants of PHEO/sPGL predisposing genes were found in 28 patients (26%). Germline pathogenic variants of *RET* were found in 29% of cases; *VHL* in 11% of cases, *NF1* in 21% of cases, *MAX* in 4% of cases, *TMEM127* in 21% of cases, *SDHB* in 7% of cases, and *SDHC* in 7% of cases (Figure 1).

Thirteen patients (46%) had multiple/bilateral PHEO/sPGL, while 15 had a unilateral PHEO. The remaining 79 tested patients were negative for germline pathogenic variants; however, 2 of these patients had a clinical diagnosis of MEN2A (PHEO and medullary thyroid carcinoma) but no germline RET pathogenic variants were identified.

### 3.2. Outcomes

Long-term follow-up data were available for 102 patients (median follow-up 14 years, range 2–43). Postoperative remission was achieved in 97% of patients; 3 patients with malignant PHEO/sPGL at diagnosis (AJCC stage III) had persistent or early recurrent disease (within 6 months) after surgery. Recurrent PHEO/sPGL was found in 15 patients (14%). The overall recurrence probability rates were 8% at 5 years, 13% at 10 years, and rose to 33% at 30 years (Figure 2).

Predictive factors of recurrence are reported in Table 2. In the univariate analysis, no significant differences concerning age at diagnosis, gender, and size of the tumor were found between recurrence-free patients and those with recurrent disease. Recurrence occurred more frequently with hereditary variants than with sporadic variants (32% vs. 9.5%, *p* = 0.03), in the presence of multiple tumors (PHEO + sPGL) at diagnosis (80% vs. 12%, *p* = 0.002) and in the presence of malignant behavior at diagnosis (67% vs. 11.5%, *p* = 0.004). In the multivariate analysis (Table 2), all of these factors remained significant independent predictors of recurrence (odds ratio 6.91, 4.14, and 79.3, respectively; *p* < 0.01).

#### 3.2.1. New Tumor Recurrence

During follow-up, 9 patients (8.6%) developed a second tumor in the contralateral adrenal gland and/or a sPGL after a mean time of 12 years (range 3–30). The 10- and 20-year new tumor recurrence rates were 7% and 14%, respectively (Figure 3a). Among the 107 patients that underwent genetic testing, a prolonged follow-up was available for 85 patients. Recurrent disease was found in 32% (7 out of 22 patients) with hereditary PHEO/sPGL and in 3% (2 out of 63) with a negative genetic test (*p* = 0.0009). The 10- and 20-year new tumor recurrence rates were significantly higher in patients with a hereditary variant than in patients with a sporadic variant (22% and 3% vs. 38% and 6.5%, respectively; *p* < 0.0001) (Figure 3b).

#### 3.2.2. Metastatic Recurrence

Metastatic recurrence after postoperative remission occurred in 6 patients (6%), including 4 patients with stage III tumors and 2 patients with stages I/II tumors. The overall 5- and 20-year metastatic recurrence rates were 5% and 8%, respectively (Figure 4a). Metastatic recurrence occurred within 5 years in the majority of patients (83%) with malignant disease (AJCC stage III) at diagnosis. The 5-year metastatic recurrence rate was significantly higher in patients with stage III tumors than in patients with stages I/II tumors (100% vs. 1%, respectively; *p* < 0.0001). Patients with apparently benign PHEO at diagnosis (AJCC stage I/II) had a 3% of risk of developing recurrence at 15 years (Figure 4b).

#### 3.2.3. Survival Analysis

Five patients died from metastatic disease (3 patients with persistent disease who died within one year and 2 patients who died 9 and 5 years after initial postoperative remission because of local and distant metastatic recurrence). The overall 5- and 10-year survival rates were 97% and 95%, respectively (Figure 5a). The 5- and 10-year survival rates were significantly lower in patients with PHEO/sPGL with malignant behavior at diagnosis than in patients with apparently benign tumors (58% and 39% vs. 100%, respectively; *p* < 0.0001) (Figure 5b). No significant difference in terms of survival rate between hereditary and sporadic PHEO/sPGL was observed (*p* = 0.46).

## 4. Discussion

PHEO and sPGL are very rare tumors with an incidence ranging from 0.04 to 0.95 cases per 100,000 per year [13]. Due to the rarity and complexity of the disease, the management and follow-up of patients with PHEO/sPGL is still debated. Most patients with PHEO/sPGL are usually lost at follow-up, since they usually remain disease-free for a long time and are considered as definitively cured, especially for those with sporadic and apparently benign disease at diagnosis. The Endocrine Society Guidelines suggest lifelong follow-up by annual biochemical testing in patients with PHEO/sPGL because of the non-negligible rates of recurrence and metastatic disease even more than 10 years after initial surgery [14]. However, the available literature is based on small single center experiences, with limited follow-up, no survival analysis, and often lacks genetic data or a clear distinction between new tumor and metastatic recurrence and between patients with malignant or benign tumor features at initial surgery, which may represent the most relevant predictive feature of recurrence. Moreover, recent reviews and meta-analyses have shown lower recurrence rates than previously reported, questioning the need of lifelong follow-up among patients with PHEO/sPGL [15,16].

The present retrospective study analyzed the long-term outcomes in terms of survival and recurrence rates in a monocentric large series of patients who underwent surgery for PHEO/sPGL, considering the presence of sporadic or hereditary variants and the behavior of the tumor at diagnosis. Moreover, the overall recurrence rate was further analyzed as new tumor recurrence (as typically may occur in genetic disease) and metastatic recurrence (as occurs in the truly malignant forms). In accordance with previous reports [11,17,18,19,20,21,22], even if most of the operated patients were cured after surgery, the overall rate of recurrent disease and PHEO/sPGL-related metastasis exceeded 15% in the present series, confirming the need for long-term follow-up. The long-term outcome analysis in this series revealed a risk of overall recurrence (either new tumor recurrence or metastases) of 13% at 10 years, which rose up to 33% at 30 years.

Several predictive factors of recurrence have been reported in the literature. In the study by Ayala-Ramirez et al. [22], a larger size and the presence of a sPGL were significantly associated with metastasis and decreased overall survival in patients with PHEO and sPGL. Eisenhofer G. et al. [23] confirmed that large tumor size and extra-adrenal location were independent risk factors for metastatic recurrence, in contrast to the findings of Goldstein et al. [17], who found no significant difference in the rate of malignancy between PHEO and sPGL.

The independent predictive factors of recurrence in our series were the presence of a hereditary variant, malignant behavior of the primary tumor at diagnosis, and the type of primary tumor (coexistence of PHEO and sPGL), while the tumor size was not a significant predictor of recurrence. As reported by Amar et al. [11], the incidence of recurrence was higher in patients with hereditary disease (33.3%). In our study, the new tumor recurrence probabilities were 7% and 14% at 10 and 20 years, respectively, and the probability was significantly higher in patients with hereditary disease than in those with sporadic disease, as expected. For this reason, a closer and tailored follow-up is usually suggested for hereditary variants, also because many different organs may be affected as part of a multisystemic syndrome. However, it should be noted that patients with a sporadic variant also had a risk, although low, of recurrence. In fact, hereditary variants cannot be excluded even with a negative genetic test, since newly discovered putative germline pathogenic variants are currently under continuous evaluation. Therefore, lifelong monitoring and follow-up is also required for sporadic cases.

Malignancy is also difficult to predict in PHEO/sPGL from histological findings. According to the WHO classification [6,7], all of these tumors should be considered at risk and theoretically malignant; metastatic disease should be defined only by the presence of metastases in non-chromaffin tissues (mainly lymph nodes and bone). In fact, although several histological algorithms have been proposed to predict malignant behavior, they have failed to reach 100% sensitivity and specificity. The most used PASS score had a positive predictive value of 31% and a negative predictive value of 99% [12]. In our series, none of the patients with a PASS score < 4 developed metastatic recurrence, while 3 of the patients with a PASS score ≥ 4 had a metastatic PHEO; thus, the PPV was 33% and NPV 100%, in line with previously reported data. Even if the 2022 WHO classification of PHEO/PGL does not entirely endorse the use of these scoring systems [6,7], it has pointed out the potential role of molecular markers and somatic pathogenic variants. For instance, the presence of somatic *SDHB* mutations is associated with the highest risk of metastatic potential [24]. Moreover, WHO experts report that mutations in *ATRX* or *SETD2*, high total somatic mutation burden, *MAML3* fusion genes, altered WNT pathway, and *TERT* activation have also been associated with increased metastatic risk [7]. Thus, the systematic search for somatic pathogenic variants in tumoral tissues might lead to a relevant predictive prognostic factor, even in truly sporadic PHEO/sPGL.

In the study by Ayala-Ramirez et al. [22], approximately 35% of the patients had metastases, frequently discovered several years after the initial diagnosis. In a review by Amar et al. [9], the reported mean incidence of malignant recurrent disease was 11.3%. In our series, metastatic recurrence after postoperative remission was found in 6% of the patients. The 5- and 20-year metastatic recurrence probabilities were 5% and 8%, respectively.

In the 2017 eighth edition of the AJCC classification [8], the terms “malignant” and “benign” PHEO were abandoned and replaced by a risk classification. Based on this staging system, stage III was defined by invasion into surrounding tissue or N positive lymph node, the prognostic value of which remains unknown. In our series, the AJCC stage III subgroup with evident malignant behavior at diagnosis (main vessel infiltration, gross adjacent tissue invasion or nodal metastases) was found in 5% of cases. Among these patients, 33% had persistent disease, 44% had recurrent metastatic disease, and 23% were still in remission. Conversely, metastatic recurrence occurred only in 2% of stage I/II patients. Additionally, in a recent multicentric study by Moog S et al. [25], locally advanced PHEO seemed to have a higher recurrence rate than the general PHEO population, requiring a closer and prolonged follow-up, including clinical, biochemical, and imaging evaluation. In fact, in these patients, the recurrence-free survival rates were 96% at 1 year and 83% at 5 years. In our study cohort, the majority of stage III subgroup patients developed distant metastatic recurrence within 5 years, with a 5-year recurrence probability of 100%. However, stage I/II patients (apparently benign tumors at presentation) also had a 3% risk of metastatic recurrence at the 15-year follow-up. Thus, the results of the present study support the 2022 WHO classification concept of considering all PHEO/sPGL cases as potentially malignant neoplasms with variable metastatic potential. Therefore, the results show that lifelong follow-up for all patients with PHEO/sPGL should be suggested, since metastatic recurrence may occur several years after the initial surgery. Locally advanced stage III PHEO is at higher risk of metastatic recurrence (100% at 5 years) and requires a more tailored and closer follow-up; however, a metastatic potential cannot be excluded even for apparently benign PHEO.

Survival studies on PHEO/sPGL are limited to small or heterogenous series. Pamporaki et al. [26] recently reported 5- and 20- year survival rates of 99% and 97%, respectively, in patients without metastases and 86% and 70%, respectively, in those with metastatic disease. In the study by Moog et al. [25], stage III patients had overall survival rates of 100% at 1 year, 95% at 5 years, and 89% at 10 years. In our cohort, 5 patients died with metastatic PHEO (3 with stage III and 2 with stage I/II at initial diagnosis). The 10-year overall survival in our study was 95%, but it fell to 39% for stage III patients.

## 5. Conclusions

All patients with PHEO/sPGL should be followed regularly after initial surgery with lifelong biochemical and imaging reassessment, even those with apparently benign and sporadic tumors, because of the risk of long-term recurrence. Hereditary and malignant variants need even stricter follow-up and can never be considered as completely cured, since long-term relapses are frequently found even if recurrence usually occurs early after initial surgery. Long-term prognosis may be poor in metastatic disease.

## Figures and Tables

**Figure 1 cancers-15-02890-f001:**
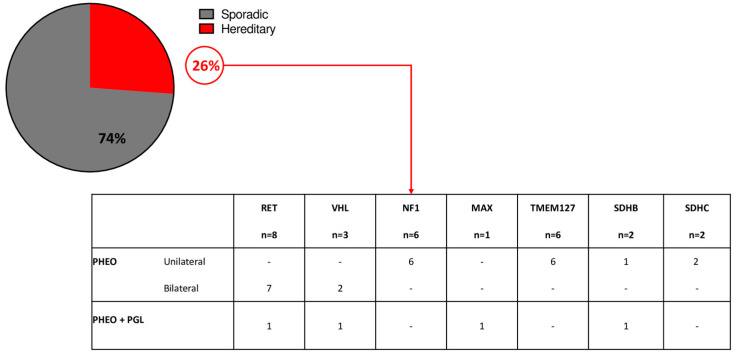
Genetic analysis in the study cohort.

**Figure 2 cancers-15-02890-f002:**
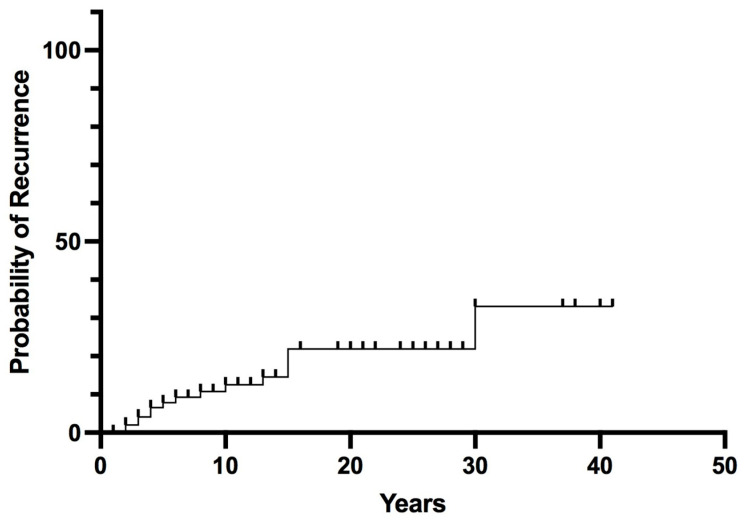
Overall Recurrence in pheochromocytoma/sympathetic paraganglioma.

**Figure 3 cancers-15-02890-f003:**
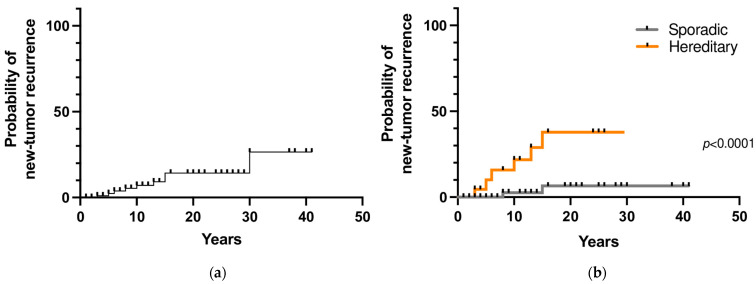
(**a**) New tumor recurrence probability in pheochromocytoma/sympathetic paraganglioma; (**b**) New tumor recurrence probability in hereditary and sporadic cases.

**Figure 4 cancers-15-02890-f004:**
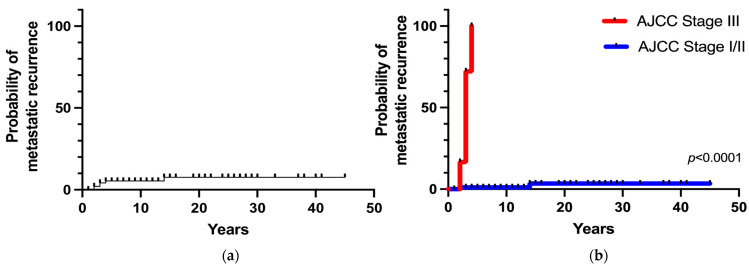
(**a**) Overall metastatic recurrence probability; (**b**) Metastatic recurrence probability according to the stage at diagnosis.

**Figure 5 cancers-15-02890-f005:**
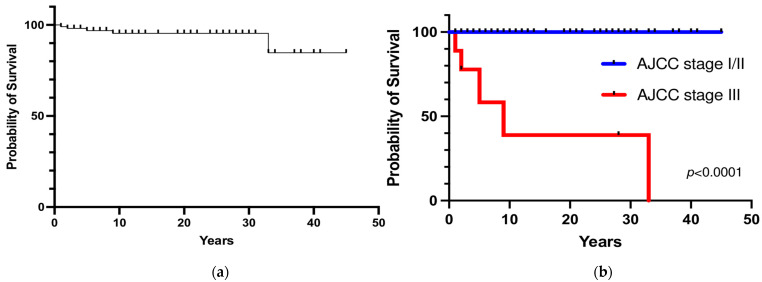
(**a**) Overall survival; (**b**) Survival rate according to the stage at diagnosis.

**Table 1 cancers-15-02890-t001:** Clinical characteristics of the 170 patients who underwent surgery for pheochromocytoma/sympathetic paraganglioma.

Clinical Features		Patients (n = 170)
Median age at first surgery		48 (6–83)
Sex ratio (F/M)		1.15
Tumor	PHEO	155 (91%)
	sPGL	10 (6%)
	PHEO + sPGL	5 (3%)
Secreting tumor	Yes	162 (95%)
	No	8 (5%)
Genetics *	Sporadic	79 (74%)
	Hereditary	28 (26%)
PASS score #	<4	37 (80%)
	≥4	9 (20%)
Tumor size (mm)	PHEO	50 (26.4)
	sPGL	54 (25.2)
Behavior at diagnosis (TNM)	Apparently benign (AJCC stage I/II)	161 (95%)
	Malignant (AJCC stage III)	9 (5%)

Data are appropriately expressed as absolute number, percentage, mean (SD), or median (range). PHEO = pheochromocytoma; sPGL = sympathetic paraganglioma; * available for 107 cases; # available for 46 cases.

**Table 2 cancers-15-02890-t002:** Univariate and multivariate analyses of predictive factors of recurrence in patients who underwent surgery for pheochromocytoma/sympathetic paraganglioma.

Factors (*n*)	Univariate Analysis			Multivariate Analysis		
	No Recurrence	Recurrence	*p*-Value	Odds Ratio	95% CI	*p*-Value
	*n* = 87	*n* = 15				
Age at first surgery (years)	47 (16)	40 (14)	0.12	-	-	-
Gender				-	-	-
Male	39	8	0.58			
Female	48	7				
Malignant at diagnosis (AJCC stage III)						
Yes	2 (2%)	4 (27%)	0.004	79.3	5.91–1070	0.001
No	85 (98%)	11 (73%)				
Variant *						
Sporadic	57 (80%)	6 (43%)	0.03	6.91	1.42–33.8	0.01
Hereditary	15 (20%)	7 (57%)				
Type of tumor						
PHEO	80 (92%)	11 (73%)	0.002	4.14	1.36–12.6	0.01
sPGL	6 (7%)	0				
PHEO + sPGL	1 (1%)	4 (27%)				
Size of the tumor (mm) PASS score #	40 (13–150)	39.5 (17–150)	0.59	-	-	-
<4	26	3	>0.99			
≥4	7	1				

Data are appropriately expressed as absolute number, percentage, mean (SD), or median (range). PHEO = pheochromocytoma; sPGL = sympathetic paraganglioma; * available for 85 cases; # available for 37 cases.

## Data Availability

Some or all datasets generated during and/or analyzed during the current study are not publicly available but are available from the corresponding author on reasonable request.

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
