# Peer review of "Long-Term Outcomes after Surgery for Pheochromocytoma and Sympathetic Paraganglioma"

_cancers, 2023, doi:10.3390/cancers15112890_

Round 1

Reviewer 1 Report

The authors present on a monocentric study of approximately 170 patients with pheochromoctyomas and paragangliomas (PPGL) reporting on germline vs sporadic tumors, pathological correlates and staging as they relate to rates of recurrence. A salient conclusion of this manuscript is that “life-long” screening is indicated in all patients with PPGL given the risk of recurrence. The reviewers should be commended for compiling a large cohort of data.This reviewer has a few comments

(1)   The WHO classification of paragangliomas and pheochromocytomas was updated in 2022 and therefore insights from this updated classification scheme must be discussed in this article. Here is an article that summarizes those updates (10.1007/s12022-022-09704-6). Notably, the updated classification specifically mentions that PPGLs are no longer considered “benign” or “malignant”; rather, all tumors must be considered malignant given the risk of recurrence. Therefore, a major conclusion of the updated classification is that all patients should undergo life-long screening as mentioned in this article. Therefore, this article provides conclusions that support these recommendations. 

(2)   Recently, it has come to light that of the known susceptibility genes for PPGL approximately 30-40% of patients have a known heritable germline pathogenic variant. However, many authorities have begun sequencing the tumors themselves. It has been found that even in “sporadic” tumors (a term here that describes tumors arising in patients without a known germline pathogenic variant) that 40-60% have a known susceptibility gene (i.e. a patient may not have a known VHL germline variant, however the tumor itself may have a known pathogenic variant in VHL). Therefore with the advent of this sequencing there is now a greater ability to correlate these somatic pathogenic variants with overall risk of recurrence and metastatic behavior/potential. Given this is a newer development in PPGL it is unrealistic that the authors in this cohort dating back to 1979 would have this information available. However, this should at least be mentioned/discussed in this article. It should be noted that even in those patients with “sporadic” tumors that further sequencing of the tumors may identify known somatic pathogenic variants and thus further inform potential for recurrence. So that readers recognize that this may* become a newer standard of care for these patients and if the option is, perhaps available, it should be performed. 

(3)   The term “genetic mutation” has now been supplanted with the term “genetic variant”. Those variants that are known to cause disease are referred to as “pathogenic variants” the language should be changed in this manuscript to reflect this. 

(4)   The manuscript needs to be carefully reviewed for English perhaps by a native speaker as the authors use a handful of idiomatic references “long-life” for example that are incorrect. The phrase is “life-long”. Additionally, the formatting needs to consistent as it is quite inconsistent (spacing between titles/paragaphs and indenting is inconsistent throughout the manuscript and distracting).

Author Response

1) We agree with and thank the reviewer for the comment. The WHO classification of PPGLs was updated in 2022; in the updated classification PPGLs are no longer considered “benign” or “malignant”; rather, all tumors must be considered malignant given the risk of recurrence. In the present paper, in fact, we classified PPGLS stage I or II (i.e <5 cm or > 5 cm, respectively, without extra-adrenal invasion, nodal or distant metastases) as “apparently benign” with indeterminate behavior, and stage III (i.e. any size with invasion into surrounding tissues and/or regional lymph node metastases) and IV (i.e. with distant metastases) as malignant. Our results demonstrated that, even if the majority of stage III patients developed distant metastatic recurrences within 5 years (5-year recurrence probability of 100%), also stage I/II patients (apparently benign tumors at presentation) had a 3% risk of metastatic recurrence at 15-year. Therefore, all PPGL should be considered potentially metastatic (malignant) and a life-long follow-up for all patients with PPGL should be suggested. Locally advanced stage III PHEO are at higher risk of metastatic recurrence (100% at 5 year) and require a more tailored and closer follow-up. 

We have now included the 2022 updated classification in the manuscript text and in references, further supporting our results and conclusions. See introduction page 2 line 53, discussion page 9 lines 311-313.

2) We agree with the reviewer. The possibility of the sequencing of the tumors may identify somatic pathogenic variants that can be correlated to a potential risk of recurrence and metastatic behavior/potential. Due to the retrospective nature of the study and the long period of observation, we were not able to have an updated somatic mutation analysis and to correlate these data with the outcome after surgery. This is an important point that should be taken into account for the next prospective studies.

We have now included this point in the discussion (see page 9 lines 281-286).

3) We agree and changed the term in text accordingly (see pages 2, 4, 5, 9).

4) The manuscript has been extensively reviewed by an English native speaker and the text has been formatted accordingly.

Reviewer 2 Report

A relevant study in an area where limited info is currently available.

I have only a few questions:

1. The Authors state that paragangliomas arise from chromaffin cells, which is usually not the case for parasympathetic paragangliomas. Since the study deals with sympathetic paragangliomas and pheochromocytomas, I would suggest either changing the title (e.g.: Long term outcome after surgery for pheochromocytoma and sympathetic paraganglioma)  or specifying that paragangliomas may arise from sympathetic or parasympathetic paraganglia, and that the cases included in the study are those arising from adrenal and extra-adrenal sympathetic paraganglia (pheochromocytoma is now considered a sympathetic paraganglioma). 

2.  Discussion and References should be updated including: 

Rindi G, Mete O, Uccella S, Basturk O, La Rosa S, Brosens LAA, Ezzat S, de Herder WW, Klimstra DS, Papotti M, Asa SL. Overview of the 2022 WHO Classification of Neuroendocrine Neoplasms. Endocr Pathol. 2022 Mar;33(1):115-154. doi: 10.1007/s12022-022-09708-2. Epub 2022 Mar 16. PMID: 35294740

Mete O, Asa SL, Gill AJ, Kimura N, de Krijger RR, Tischler A. Overview of the 2022 WHO Classification of Paragangliomas and Pheochromocytomas. Endocr Pathol. 2022 Mar;33(1):90-114. doi: 10.1007/s12022-022-09704-6. Epub 2022 Mar 13. PMID: 35285002

Gimenez-Roqueplo AP, Robledo M, Dahia PLM. Update on the genetics of paragangliomas. Endocr Relat Cancer. 2023 Mar 8;30(4):e220373. doi: 10.1530/ERC-22-0373. PMID: 36748842; PMCID: PMC10029328

Finally, English should be improved, particularly in the Abstract. These are minor changes, overall, the text is quite well-written and easily understandable

Author Response

  1. We agree to the reviewer and we changed the title and the text accordingly (see the corrections highlighted in red in the main text).
  2. We agree and we included the references in the text (see reference 4, 6,7).
  3. The manuscript has been extensively reviewed by an English native speaker.

Round 2

Reviewer 1 Report

This is an appropriate revision in light of reviewer comments. A handful of remaining comments 

1. The english still requires revision, in reading through this version I have highlighted areas that require revision. 

2. Be consistent with the use of hyphens i.e. (life-long vs life long vs lifelong and follow-up vs follow up) 

3. Again "long-life" is not the correct idiomatic phrase it is "life-long" this should be consistently corrected throughout the manuscript. 

Author Response

Reviewer#1

This is an appropriate revision in light of reviewer comments. A handful of remaining comments 

1)The english still requires revision, in reading through this version I have highlighted areas that require revision.

Response: We would like to thank the reviewer for the careful revision of our work. The manuscript has been revised following the suggestions of the reviewer (see the manuscript text; we have highlighted in blue the changes to ease their tracking).

2. Be consistent with the use of hyphens i.e. (life-long vs life long vs lifelong and follow-up vs follow up).

Response: we have modified the text accordingly.

3. Again "long-life" is not the correct idiomatic phrase it is "life-long" this should be consistently corrected throughout the manuscript. 

Response: we have modified the text accordingly.